# Long-Term Application of Biochar Mitigates Negative Plant–Soil Feedback by Shaping Arbuscular Mycorrhizal Fungi and Fungal Pathogens

**DOI:** 10.3390/microorganisms12040810

**Published:** 2024-04-17

**Authors:** Mohamed Idbella, Silvia Baronti, Francesco Primo Vaccari, Ahmed M. Abd-ElGawad, Giuliano Bonanomi

**Affiliations:** 1College for Sustainable Agriculture and Environmental Sciences, Mohammed VI Polytechnic University, Ben Guerir 43150, Morocco; 2Institute of BioEconomy (IBE), National Research Council (CNR), Via Giovanni Caproni, 8, 50144 Firenze, Italy; silvia.baronti@ibe.cnr.it (S.B.); francesco.vaccari@ibe.cnr.it (F.P.V.); 3Plant Production Department, College of Food & Agriculture Sciences, King Saud University, P.O. Box 2460, Riyadh 11451, Saudi Arabia; aibrahim2@ksu.edu.sa; 4Department of Agricultural Sciences, University of Naples Federico II, Via Università 100, 80055 Portici, Italy; giuliano.bonanomi@unina.it; 5Task Force on Microbiome Studies, University of Naples Federico II, 80138 Naples, Italy

**Keywords:** plant–soil feedback, fungal pathogens, AMF, *Vitis vinifera*, biochar, next-generation sequencing

## Abstract

Negative plant–soil feedback (PSF) arises when localized accumulations of pathogens reduce the growth of conspecifics, whereas positive PSF can occur due to the emergence of mutualists. Biochar, a carbon-rich material produced by the pyrolysis of organic matter, has been shown to modulate soil microbial communities by altering their abundance, diversity, and activity. For this reason, to assess the long-term impact of biochar on soil microbiome dynamics and subsequent plant performance, we conducted a PSF greenhouse experiment using field soil conditioned over 10 years with *Vitis vinifera* (L.), without (e.g., C) or with biochar at two rates (e.g., B and BB). Subsequently, the conditioned soil was employed in a response phase involving either the same plant species or different species, i.e., *Medicago sativa* (L.), *Lolium perenne* (L.), and *Solanum lycopersicum* (L.). We utilized next-generation sequencing to assess the abundance and diversity of fungal pathogens and arbuscular mycorrhizal fungi (AMF) within each conditioned soil. Our findings demonstrate that biochar application exerted a stimulatory effect on the growth of both conspecifics and heterospecifics. In addition, our results show that untreated soils had a higher abundance of grape-specialized fungal pathogens, mainly *Ilyonectria liriodendra*, with a relative abundance of 20.6% compared to 2.1% and 5.1% in B and BB, respectively. *Cryptovalsa ampelina* also demonstrated higher prevalence in untreated soils, accounting for 4.3% compared to 0.4% in B and 0.1% in BB. Additionally, *Phaeoacremonium iranianum* was exclusively present in untreated soils, comprising 12.2% of the pathogens’ population. Conversely, the application of biochar reduced generalist fungal pathogens. For instance, *Plenodomus biglobosus* decreased from 10.5% in C to 7.1% in B and 2.3% in BB, while *Ilyonectria mors-panacis* declined from 5.8% in C to 0.5% in B and 0.2% in BB. Furthermore, biochar application was found to enrich the AMF community. Notably, certain species like *Funneliformis geosporum* exhibited increased relative abundance in biochar-treated soils, reaching 46.8% in B and 70.3% in BB, compared to 40.5% in untreated soils. Concurrently, other AMF species, namely *Rhizophagus irregularis*, *Rhizophagus diaphanus*, and *Claroideoglomus drummondii*, were exclusively observed in soils where biochar was applied. We propose that the alleviation of negative PSF can be attributed to the positive influence of AMF in the absence of strong inhibition by pathogens. In conclusion, our study underscores the potential of biochar application as a strategic agricultural practice for promoting sustainable soil management over the long term.

## 1. Introduction

Plant–soil feedback (PSF) encompasses a complex set of interactions through which a particular plant species shapes the composition of its own soil biotic and abiotic properties, subsequently influencing the establishment and growth of both conspecific and heterospecific plants in that soil [1,2]. The strength and direction of PSF can have a significant impact on community composition, diversity, and ecosystem functioning [3]. Negative PSF occurs when locally accumulated pathogens suppress the growth of conspecific individuals, whereas positive PSF arises from the establishment of beneficial microbial mutualists, such as mycorrhizal fungi [4]. It is important to note that the direction of PSF depends on several factors, including the type of mycorrhizal association involved. Numerous studies have shown that invasive plant species can establish themselves in natural habitats by managing to escape their native antagonists [5,6]. However, there is compelling evidence suggesting that when heterospecific crops are cultivated in monocultures, they can form species-specific associations with soil-borne pathogens already present in the soil, leading to negative PSF rather than the anticipated positive outcomes [7]. Alternatively, recent research has highlighted the significant role of specific mycorrhizal fungal species in determining the direction and strength of PSFs [8,9]. Negative feedback is generally associated with plant species forming arbuscular mycorrhizal fungi (AMF) associations, while positive feedback is more commonly observed in plant species engaging in ectomycorrhizal associations [10]. Thus, understanding the factors that govern the direction of PSFs necessitates a comprehensive understanding of how both mutualists and pathogens independently influence the feedback process.

Biochar is a product resulting from thermal pyrolysis of organic materials under oxygen-deprived, high-temperature conditions [11]. Its application in soil management has garnered considerable scientific interest due to its noteworthy impacts on carbon sequestration, soil fertility enhancement, and overall soil quality amelioration [12,13,14]. Importantly, the incorporation of biochar into soil has been demonstrated to reduce nutrient leaching [15], reduce soil bulk density, augment water retention, enhance cation exchange capacity, and modulate pH levels [16]. Furthermore, biochar exhibits proficiency in the sequestration and alleviation of soil contaminants [17,18]. Its inherent recalcitrance allows biochar to persist within the soil for a long period, setting it apart from crop residues or animal manure, which undergo relative degradation over time [19]. Biochar engenders a spectrum of effects on soil properties, soil microbiota [20], rhizosphere microbiome [21], plant growth, and crop yields [22], while also conferring enhanced plant resistance to diseases [23,24,25]. The diverse responses of soil microbial biomass to biochar application have been documented in numerous studies, encompassing increases [26], decreases [27], and neutral effects [28,29]. These responses are contingent upon several factors, including biochar characteristics (e.g., feedstock, nutrient composition, and pyrolysis temperature), initial soil conditions, land-use practices, management strategies, and plant species functional type. Additionally, biochar is recognized for its capacity to reshape soil bacterial community structure and diversity [30,31]. Furthermore, investigations have probed the influence of biochar on fungal abundance and diversity, revealing a range of outcomes, including fluctuations in the colonization and abundance of AMF [32,33], diminished fungal diversity attributed to the inability of certain fungal taxa to adapt to rapid soil environmental changes [31], and a reduction in fungal abundance [34]. Moreover, biochar has been observed to mitigate disease incidence and severity caused by fungal pathogens, largely attributable to its modification of soil microbiota and induction of plant systemic resistance [35,36]. Therefore, it could be hypothesized that biochar application can be considered an ecological practice for mitigating the negative PSF caused by soil-borne pathogens in agroecosystems.

The predominant method for investigating microbial influence on negative plant–soil feedback (PSF) has thus far been the ‘soil history’ approach [37]. This approach involves transferring small portions of pre-conditioned soil, serving as a microbial inoculum, into sterilized soil. Subsequently, the performance of plants is assessed during the response phase. This approach offers the advantage of emphasizing the crucial role of microbes in shaping PSF. However, it is essential to recognize that in natural settings, soil experiences concurrent effects such as nutrient depletion and the release of chemicals from decomposing leaf and root litter. Furthermore, many reported PSF experiments are typically short-term, resulting in a limited conditioning period and the minimal impact of plant conditioning on the soil. To address this limitation, we embarked on a long-term field experiment spanning more than a decade. In this experiment, soil underwent conditioning through monoculture cultivation of *Vitis vinifera* (L.) for multiple cycles. This soil was part of a long-term field study, initiated in 2009, aiming to assess the effects of biochar application in one or two doses on soil hydrology, plant physiology, crop yields, and grapevine quality [38]. This conditioned soil was subsequently employed in a response phase, featuring the same plant species, *V. vinifera*, or different plant species including *Medicago sativa* (L.), *Lolium perenne* (L.), and *Solanum lycopersicum* (L.). We chose *V. vinifera* to investigate the impacts of monoculture and biochar application on the soil’s suitability for replanting the same species, considering the relevance of the replant disease issue for various orchard species and grapevines. *S. lycopersicon*, *M. sativa*, and *L. perenne* were selected to capture diverse plant responses, reflecting their distinct physiological traits and ecological roles. Subsequently, we conducted next-generation sequencing to analyze the abundance and diversity of fungal pathogens and AMF in each of the conditioned soils. We expected that *V. vinifera* would experience a greater negative PSF compared to heterospecific plants. This effect was hypothesized to result from the selective accumulation of pathogens relative to beneficial microbes. In addition, we hypothesized that the application of biochar would help alleviate the negative PSF for *V. vinifera* throughout its well-established effects on soil microbial communities, probably by reducing fungal pathogens and enhancing the AMF community in the soil.

## 2. Materials and Methods

The experiment consisted of two phases: the so-called conditioning phase and the response phase. The conditioning phase referred to the growth of *V. vinifera* in a monoculture in the soil of a long-term field experiment with the applied treatments, and the response phase was carried out using the conspecific species, i.e., *V. vinifera*, and the heterospecifics, i.e., *M. sativa*, *L. perenne* and *S. lycopersicum*.

### 2.1. Conditioning Phase and Biochar Application

Within the vineyard “La Braccesca Estate” (Marchesi Antinori srl, Florence, Italy) in Montepulciano (Tuscany, Central Italy, 43°10’15″ N, 11° 57’43″ E, 290 m a.s.l.), a plot trial with three treatments and five replicates was carried out in 2009. Each of the 15 plots measures 225 m^2^, with dimensions of 7.5 m in width and 30 m in length. This area includes 4 vineyard rows and 3 intermediate rows. The treatments were a single biochar application of 22 tons ha^−1^ in 2009 (B), two biochar applications of 22 tons ha^−1^ each in 2009 and 2010 (BB), and untreated control plots (C).

The vineyard (cv. Merlot, clone 181; rootstock 3309 Couderc) was established in 1995, and features rows of plants-oriented east–west, with the spaces between them partially covered by natural grass. It relies solely on natural precipitation, foregoing irrigation. Twice a year, it receives a dose of NPK fertilizer (15.0.26) at a rate of 120 kg ha^−1^. Situated in a Mediterranean climate, the region experiences an average annual temperature of 14.6 °C and accumulates 776 mm of rainfall between 2009 and 2019. The soil, classified as acidic and possessing a sandy–clay–loam texture according to USDA standards (2005), consists of 35% clay, 20% silt, and 45% sand. Compaction is significant below a depth of 0.4 m.

The biochar used in the experiment is commercially available (Bagnacavallo, Ravenna, Italy) and was obtained from orchard pruning feedstock; it was made at a low temperature (500 °C) via a slow pyrolysis process and had 77.8% OC, 0.91% N, 101 cmol(+) kg^−1^ CEC, and 2722 mm^3^ g^−1^ porosity. Biochar was applied superficially to the soil between the vineyard rows using a spreader and then mechanically incorporated into the soil to a depth of 30 cm using a chisel plough tiller. The biochar had a water content of 25%, resulting in each application corresponding to 16.5 tons of dry biochar per hectare. The vineyard operates on a three-year rotational management system. Each year, one specific inter-row is tended to by the farm, employing a rototiller and ploughing to a depth of 0–20 cm. Meanwhile, the two adjacent inter-rows remain uncultivated and are naturally covered with volunteer grass, which is mowed twice annually. The soil of the experimental vineyard is described in the work of Baronti et al. [38] and Genesio et al. [39], and the biochar properties are described in Table 1.

After 10 years, soil samples for the response phase were collected for each treatment from three types of conditioned soils: soils conditioned with *V. vinifera* without treatment, with the application of biochar in one dose, and with the application of biochar in two doses. For further analysis, soil samples were collected using a 10 cm diameter soil corer, reaching a depth of 0–20 cm after the removal of above-ground litter. This process was performed at three distinct points within each replicate plot to create a composite sample representative of each plot. A total of five replicate samples were collected for each treatment, resulting in 15 samples overall. Subsequently, soil samples were processed in the laboratory. Initially, they were sieved at field moisture using a 2 mm mesh. Following sieving, the samples were divided into three portions: one portion was stored at 4 °C for the analysis of soil biochemical activities, another portion was preserved at −80 °C for DNA extraction, and the remaining portion was air-dried for subsequent chemical analysis. The chemical, biochemical, and metagenomics properties of each of the conditioned soils are described in the work of Idbella et al. [40].

### 2.2. Response Phase

In this phase, conducted in a greenhouse of the Department of Agriculture (Federico II University of Naples, Italy), the pots (20 cm opening diameter × 18 cm height × 15 cm bottom diameter) were filled with the previously conditioned soils, which included soil with a single biochar application, soil with two biochar applications, and a control without biochar. Each treatment, as well as each of the tested plant species, had 5 replicates, resulting in a total of 60 pots (4 plant species × 3 treatments × 5 replicates). For each heterospecific plant species, ten seeds were sown in each pot. These seeds underwent surface sterilization in a 3% sodium hypochlorite solution for 1 min and were thoroughly rinsed with sterile water before use. After germination, the number of seedlings in each pot was reduced to five. In the case of *V. vinifera*, two seedlings of the same variety (cv. Merlot, clone 181; rootstock 3309 Couderc) were placed in each corresponding pot. Pots were arranged in a randomized block design, and each pot was located inside an individual pot saucer in order to avoid contamination among plants through leaching or splashing when watering. Pots were watered to field capacity, when necessary, by adding distilled water to individual containers. Following the response phase, which lasted for 100 days for *S. lycopersicum*, 70 days for *M. sativa*, 120 days for *L. perenne*, and 230 days for *V. vinifera*, all plants survived and were harvested. Plants were cut at ground level, and the shoots were subjected to drying at 70 °C for 72 h, after which their dry weight was recorded.

### 2.3. Soil Chemical and Microbial Properties

The soil chemical and microbiota properties of the three conditioned soils were previously reported by Idbella et al. [40]. Here, the microbiota data were used as a reference dataset to investigate fungal functionality using FUNGuild analysis [41]. From the identified putative fungal functional groups/guilds, the abundances of pathogens and AMF were extracted and elaborated. For chemical analysis, soil samples were analyzed as follows. Organic carbon was determined by wet oxidation of SOM followed by titration of excess dichromate. Nitrate concentration was determined by extraction in distilled water followed by quantification by ion chromatography (Dionex, Sunnyvale, CA, USA, DX120). Ammonium was extracted from soil with KCl and quantified colorimetrically. Available P was extracted and quantified by UV spectrophotometry at 880 nm. The pH was measured in a soil-distilled water solution using a probe with selective sensors (XS Instruments, Carpi, MO, Italy). The bulk density of the upper layer was determined by the core method using metal cylinders. Samples were then weighed at field conditions, oven dried at 105 °C for 48 h, and reweighed to calculate moisture content. Bulk density was calculated as the ratio between the dry weight and the volume of the sample.

Regarding the soil microbiome, next-generation sequencing data of the fungal microbiota were analyzed using ITS gene sequences and are available in the NCBI Sequence Read Archive (SRA) under the “Braccesca & Biochar” bio-project with the accession number PRJNA842337. In our previous work [40], we found no significant differences in the Shannon diversity index, whereas the fungal phyla showed significant variation between treatments. Briefly, the Basidiomycota phylum showed an increasing range correlated with biochar rate, starting from 9.4% in C soil to 17.0% in B soil and 24.8% in BB soil. However, the Ascomycota phylum showed a decreasing range with increasing biochar rate, with a relative abundance of 73.8% in C soil, 64.8% in B soil and 58.7% in BB soil. On the other hand, Glomeromycota was more abundant in the rhizosphere of the C soil compared to the other soils, while Chytridiomycota was more abundant in the biochar soils and almost absent in the control soil.

### 2.4. Data Analysis

The statistical significance of the biomass data obtained from the experiment was evaluated using a two-way ANOVA (analysis of variance) to determine the main and interactive effects of the fixed factors of conditioning treatment and plant species on shoot biomass. The results of the analysis of variance were submitted by pairwise Tukey test comparing the individual means of response plants in each soil treatment. The level of significant differences was evaluated with *p* < 0.05. All statistical analyses were performed using STATISTICA software (Version 13.3.0, TIBCO Software Inc., Palo Alto, CA, USA), which offers a comprehensive suite of tools for data exploration, visualization, and hypothesis testing.

Both fungal pathogens and AMF relative abundance matrices were generated based on FUNGuild analysis. Furthermore, the Bray–Curtis dissimilarity between each pair of samples in each matrix was calculated. The spatial autocorrelation as well as the correlation in the composition of the two matrices were calculated using the Mantel test. In addition, co-occurrence network analyses were performed for the two guilds in the three different soils. Pairwise correlations between taxa were calculated using Spearman correlation in R (Hmisc package 4.0–1). Based on statistical analysis, only strong and significant (Spearman’s r > 0.6 or r < −0.6 and *p* < 0.05) correlations were considered. The network was visualized using Gephi (version 0.9.2) [42] to compare the co-occurrence of fungal pathogens and AMF. On the other hand, the core microbiome was identified for both pathogens and AMF by generating Venn diagrams for 3 sets (i.e., 3 treatments) using R software (Version 4.0.4) and the VennDiagram package (version 2.12.1) [43].

Based on a resemblance matrix calculated using Bray–Curtis dissimilarity, non-metric multidimensional scaling (NMDS) analyses based on the abundance of the two matrices (pathogens and AMF) were performed using Past software (version 4). The vector fit of the environmental variables to the NMDS ordination was determined using six principal components of the chemical traits. The significance of compositional changes between communities in each of the two matrices was tested using PERMANOVA (999 permutations) with soil treatment as a fixed factor.

## 3. Results

### 3.1. Chemical Properties of the Conditioned Soils

Our results showed significant changes in soil properties after the introduction of biochar (Table 2). The pH values showed an upward trend, increasing from 6.33 in C to 6.83 and 7.07 in treatments B and BB, respectively. The organic carbon content also increased significantly from 12.7 g·kg^−1^ in C to 17.3 g·kg^−1^ in B and 23.1 g·kg^−1^ in BB. In contrast, the addition of biochar reduced the bulk density from 1.63 g·cm^−3^ in C to 1.59 g·cm^−3^ and 1.53 g·cm^−3^ in treatments B and BB, respectively. The nitrate content increased with the addition of biochar from 1.51 mg·kg^−1^ in C to 1.78 mg·kg^−1^ in B and 5.88 mg·kg^−1^ in BB. Similarly, the ammonium concentration increased from 12.2 mg·kg^−1^ in C to 13.4 mg·kg^−1^ in B and 14.3 mg·kg^−1^ in BB. Finally, the phosphorus content increased significantly, from 147.4 mg·kg^−1^ in C to 262 mg·kg^−1^ in B and 313 mg·kg^−1^ in BB.

### 3.2. Plants’ Performances in the Response Phase

In both conspecific and heterospecific plants grown in soil previously conditioned with *V. vinifera*, shoot biomass was significantly increased by the addition of biochar in one dose (B) compared with the control (C) (Figure 1). For *S. lycopersicum* and *L. perenne*, the addition of biochar in two doses (BB) showed no significant difference compared to the control (C).

### 3.3. Fungal Pathogens and AMF Community Composition 

At the species level, fungal pathogens and AMF communities differed between treatments (Figure 2). Specifically, the control soil (C) was characterized by a higher abundance of generalist pathogens such as *Plenodomus biglobosus* (with 10.5% compared to 7.1% in B and 2.3% in BB) and *Ilyonectria mors-panacis* (with 5.8% compared to 0.5% in B and 0.2% in BB) as well as a higher abundance of grape-specialized pathogens such as *Ilyonectria liriodendra* (with 20.6% compared to 2.1% in B and 5.1% in BB), *Cryptovalsa ampelina* (with 4.3% compared to 0.4% in B and 0.1% in BB), and *Phaeoacremonium iranianum* (which was present only in treatment C, with 12.2%, and absent in the other treatments). On the other hand, other general pathogens like *Ilyonectria macrodidyma* and *Fusarium solani* were more abundant in BB with 19.9% and 9.7%, respectively, compared to 4.6% and 1.3% in C and 5.6% and 4.4% in B, respectively. *Fusarium acutatum*, however, was abundant in all soils but was more common in BB (56.2%), followed by B (53.5%), and it accounted for 32.7% in the control. *Boeremia exigua*, *Acremonium persicinum*, and *Acremonium furcatum* were more abundant in soil B (with 6.3%, 8.9%, and 3.8%) than in soil C (with 2.6%, 1.0%, and 0%) and soil BB (with 1.1%, 0.1%, and 2.5%, respectively).

Regarding AMF, *Funneliformis geosporum* and *Paraglomus laccatum* were common in all soils, but *F. geosporum* was more abundant in BB with 70.3%, followed by B soil with 46.8% and C soil with 40.5%; *P. laccatum* was much more abundant in the control soil with 60.2%, followed by B soil with 20.5% and BB soil with 18.8%. *Rhizophagus irregularis* and *Rhizophagus diaphanus*, on the other hand, were not present in C soil, but were abundant in B and BB soils with 20.4% and 5.7%, respectively. Finally, *Claroideoglomus drummondii* was present only in B soil, with a relative abundance of 9.8%.

Correlation analysis showed that the relative abundance and richness of fungal pathogens had a strong, significant positive correlation with the relative abundance and richness of AMF (r = 0.21, *p* = 0.035) (Figure 3A). Moreover, our analysis of the co-occurrence network shows that among the correlations between fungal pathogens and AMF communities, only positive interactions were strong and significant (Figure 3B). Overall, 36 species were determined to be the core pathogen species of the studied microbiomes. The Venn diagram (Figure 3C) confirms that all treatments contain a high number of unique pathogen species, ranging from 17 in BB to 16 in C and 13 in B. The core AMF species were four, and the number of unique AMF species ranged from five in BB to four in C and only one in B.

### 3.4. Linking Pathogens and AMF to Soil Chemical Traits

The NMDS analysis shows the relationships between fungal pathogen and AMF communities with soil chemical properties (Figure 4). We found that the ordination of both pathogens and AMF in B soil was strongly correlated with pH, P, ammonium, and organic carbon content, while the ordination of BB samples was correlated with nitrate content and the ordination of C samples was correlated with soil bulk density.

## 4. Discussion

Our study involved a comparison of the responses of *V. vinifera* cuttings and three herbaceous plants in soil that had been conditioned for 10 years with *V. vinifera*, both with and without biochar. The results demonstrated that biochar had a positive impact on the growth of both conspecific and heterospecific plants, particularly in mitigating the negative PSF observed in grape cultivation. This aligns with the findings of Wang et al. [44], who observed that biochar holds potential for alleviating negative PSF by modifying the soil microbiome. Specifically, it enriched beneficial bacteria with antagonistic activity against pathogens, thereby enhancing the sustainability of soil in agricultural systems, such as the cultivation of sanqi (*Panax notoginseng*). Biochar is recognized for the significant improvements it makes to soil’s physical properties. Its application enhances soil aggregate stability and water-holding capacity by improving soil pore characteristics and water retention [45]. Furthermore, biochar enhances soil fertility by facilitating the biochemical cycling of nitrogen and phosphorus [46]. The organic matter and inorganic ions present in biochar also provide essential nutrients to plants, which may explain the enhanced growth observed in our study. However, our results indicated that increasing the amount of biochar caused no significant changes in most cases when compared to a single application. In some instances, plant growth even decreased to levels approximately equal to the control. Similarly, a meta-analysis by Jeffery et al. [47] revealed no statistically significant differences between various application rates, with rates of 10, 25, 50, and 100 t ha^−1^ all significantly improving crop productivity compared to controls without biochar. Additionally, high biochar application rates, as reported by Mia et al. [48], can negatively impact plant growth due to salt stress resulting from the high electrical conductivity of biochar. Likewise, Zhang et al. [49] applied biochar at rates of 20 and 40 t ha^−1^ for corn cultivation, reporting a 7–14% growth increase compared to untreated areas, with the 20 t ha^−1^ application rate yielding higher growth rates.

Our results reveal that untreated soils exhibited a higher abundance of specialized fungal pathogens specific to grapes when compared to biochar-treated soils, notably *Ilyonectria liriodendra* and *Cryptovalsa ampelina*. Furthermore, *Phaeoacremonium iranianum* was exclusively present in the untreated soils. These fungal pathogens are known to selectively target grapes and can lead to severe diseases, including the “esca” complex [50]. For instance, *I. liriodendra* is a recognized primary causative agent of black foot disease in grapevines, and a similar prevalence of this species has been documented in infected vineyards across various regions, such as Australia [51], California [52], Chile [53], New Zealand and South Africa [54], Portugal [55], eastern Canada [56], Spain [57], Uruguay [58], and Turkey [59]. Additionally, Mostert et al. [60] isolated a diatrypacic ascomycete identified as *C. ampelina* from grapevines in South Africa and Australia, with pathogenicity confirmed through wound inoculation. *C. ampelina* was also reported in *V. vinifera* in northeastern Spain, with its pathogenicity to grapevines confirmed, although the authors suspected a moderate level of virulence for this fungus [61]. On the other hand, Gramaje et al. [62] demonstrated that *P. iranianum* is one of the main fungi isolated from rootstocks of young grapevines displaying symptoms of Petri’s disease in Spain, including low vigor, reduced foliage, and dark streaking in the xylem. Nevertheless, our results indicate that biochar application can also reduce the incidence of other soil-borne fungal pathogens that do not selectively target grapes. For example, *Plenodomus biglobosus*, a significant pathogen responsible for causing Phoma stem canker in Brassicas, including *B. napus* (oilseed rape) [63], and in *Eutrema japonicum* (wasabi) [64], was mitigated by biochar application. Similarly, *Ilyonectria mors-panacis*, a major plant pathogen responsible for ginseng root rot [65], was also diminished through biochar application. In fact, biochar has been shown to be beneficial for plant health, with concentrations of up to 3% reducing various foliar [35,66] and root pathogens [33,67,68]. The mechanism behind this action likely involves an induced systemic response in the plant [66]. Therefore, our results suggest that the reduction in negative PSF associated with biochar addition may be attributed to the suppression and reduced abundance of grapevine-specific pathogens.

Furthermore, our results also provide insights into the enrichment of the AMF community through the addition of biochar. We observed that *F. geosporum* was more abundant in the amended soils than in the control soils, and *R. irregularis*, *R. diaphanus*, and *C. drummondii* were exclusively present in the biochar-amended soils. Several studies have demonstrated that biochar addition enhances AMF colonization rates [69,70]. In general, biochar application can directly impact soil microbes, such as AMF [69,71], through the nutrients contained within the biochar itself [72,73]. It can also indirectly affect them by absorbing and binding nutrients [20], which subsequently influences plant growth. Using 33P, Hammer et al. [74] confirmed that AMF hyphae can penetrate the micropores of biochar to extract phosphorus. The formation of a hyphal network by AMF with plant roots significantly enhances root access to a large soil surface area, leading to improved plant growth [75]. AMF enhance plant nutrition by increasing the availability and translocation of various nutrients [76]. Furthermore, AMF improve soil quality by influencing its structure and texture, thus contributing to plant health [77]. AMF also enhance the water regime in the soil and increase a plant’s tolerance to drought, temperature extremes, heavy metals, salinity, and metal pollution [78]. Therefore, increasing the abundance of AMF in the soil by adding biochar would amplify their positive effects and, in turn, improve plant growth by mitigating the negative PSF. Conversely, the observed reduction in negative PSF could be attributed to the known ability of AMF to influence soil-borne pathogens [68,79], suggesting that biochar contributed to an increase in the abundance of AMF, thereby enhancing their capacity to control disease incidence in amended soils. In fact, AMF are recognized to have a suppressive effect on soil-borne diseases, and increased root colonization improves disease suppression [68]. Previous studies have reported a reduction in Fusarium root rot in AMF-colonized asparagus plants in biochar-enriched soils [33,80]. However, our correlation analysis revealed a strong, significant positive correlation between the relative abundance and richness of fungal pathogens and the relative abundance and richness of AMF. Additionally, our co-occurrence network analysis showed only significant positive interactions between AMF and fungal pathogens, ruling out the possibility of antagonism and a disease-suppressive effect of AMF. On the contrary, AMF can stimulate changes in various compounds in plants infected by pathogens, such as nitric oxide [81], abscisic acid [82], jasmonic acid [83], salicylic acid [84], gibberellin [85], and strigolactone [86], thereby triggering a defense response in plants rather than a direct interaction with pathogens.

Our nMDS analysis unveiled a multifaceted interaction among parameters shaping the distribution of fungal communities. Within both B and BB soils, a positive correlation was observed between fungi and parameters such as P, pH, OC, ammonium, and nitrate concentrations. These associations imply that the incorporation of biochar into soils fosters conditions conducive to fungal proliferation, characterized by enhanced nutrient availability, pH levels, and increased carbon content [87,88]. Notably, soil pH serves as a reflection of soil nutrient availability and plays a pivotal role in regulating the accessibility of other nutrients via ion exchange processes [89]. The long-term increase in soil pH values from 6.33 in C to 6.83 in B and 7.07 in BB, observed ten years after biochar application, persists even under conventional vineyard management. This elevation may be attributed not solely to the initially high pH value (9.8) of the biochar utilized [39], but also to the presence of base cations in the soil and the enduring effects of biochar basic oxides [90]. Additional mechanisms contributing to pH elevation could involve processes associated with plant-mediated NO_3_-N sorption, which can generate OH^−^ ions and subsequently raise pH levels. This phenomenon contrasts with NH_4_^+^ uptake, which yields H^+^ ions, thereby contributing to soil acidification [91]. Capillary forces within biochar micropores possess the capacity to mitigate the acidifying potential of such ions within the soil solution. Upon contact with biochar micropores, these ions may experience reduced diffusion and adsorption onto biochar surfaces due to capillary effects. Consequently, the immediate availability of ions for reactions leading to soil acidification is diminished. The observed increase in soil pH in the long term suggests improved conditions for grapevine cultivation, potentially enhancing nutrient uptake and water retention. Moreover, the mitigation of soil acidification by biochar may contribute to the sustainability of vineyard ecosystems, preserving soil fertility and minimizing the need for external amendments. Moreover, fluctuations in soil pH can precipitate alterations in species richness and community composition. Specifically, soil pH exerts significant influence on the physiological dynamics of AMF, affecting crucial aspects such as sporulation [92] and hyphal growth [93]. Therefore, the observed shifts in AMF physiology induced by changes in soil pH, facilitated by biochar application, may contribute to the reduction of fungal pathogen populations in agricultural soils. This underscores the multifaceted implications of soil pH modulation through biochar amendments, not only on fungal community dynamics but also on plant health and disease suppression in agroecosystems.

## 5. Conclusions

Our study, which investigated (for the first time) 10 years of conditioning with biochar at two different rates, offers valuable insights into the potential of biochar to act as a sustainable soil management tool in agricultural ecosystems. By conducting a feedback experiment over a decade-long period, we demonstrated that biochar amendment can effectively mitigate the negative PSF effects associated with continuous cropping of *V. vinifera*. Our results not only underscore the importance of biochar in enhancing plant growth, but also shed light on its role in modulating soil microbial communities, particularly AMF and fungal pathogens. The practical applications of our findings extend beyond improving crop productivity to encompass environmental sustainability and resilience in agricultural systems. By reducing the relative abundance of specialized fungal pathogens and enhancing the diversity of beneficial AMF, biochar application can contribute to the development of more resilient agroecosystems. Furthermore, the absence of antagonism between soil communities suggests that biochar promotes a balanced soil microbiome conducive to plant growth. However, several questions remain unanswered, presenting avenues for future research. Firstly, further investigation is warranted to elucidate the underlying mechanisms through which biochar interacts with soil microbial communities and influences plant–soil interactions. Additionally, long-term field studies are needed to assess the scalability and durability of biochar-mediated improvements in soil health and crop performance across diverse agroecological contexts. Moreover, exploring the potential synergies between biochar and other soil amendments or management practices could enhance our understanding of integrated soil fertility management strategies.

## Figures and Tables

**Figure 1 microorganisms-12-00810-f001:**
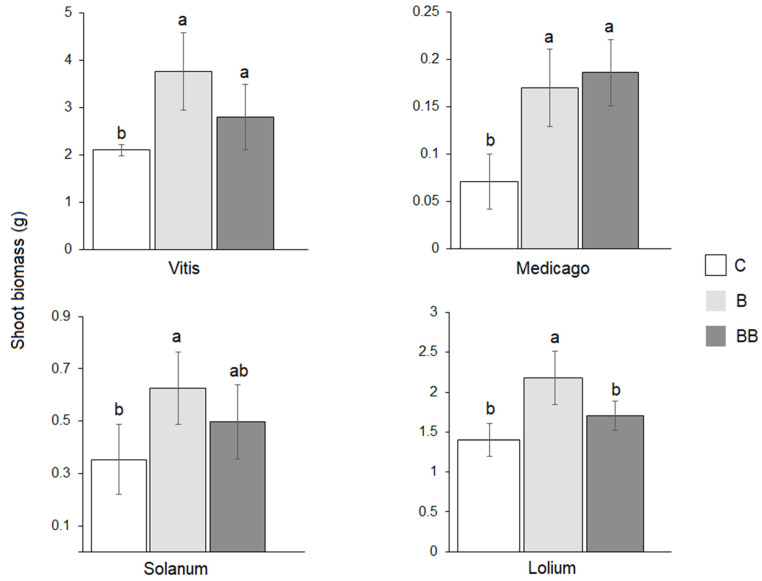
Shoot biomass (g. plant^−1^) of *Vitis vinifera* L., *Medicago sativa* L., *Solanum lycopersicum* L., and *Lolium perenne* L. under three different treatments: control (C), biochar (B), and double application of the biochar (BB). The bar plots represent the response of the *V. vinifera* plants in the conditioned soil after the same treatments. The error bars represent the standard deviation for each plant species. Bars topped by the same letter do not differ significantly according to Tukey’s test on each plant species.

**Figure 2 microorganisms-12-00810-f002:**
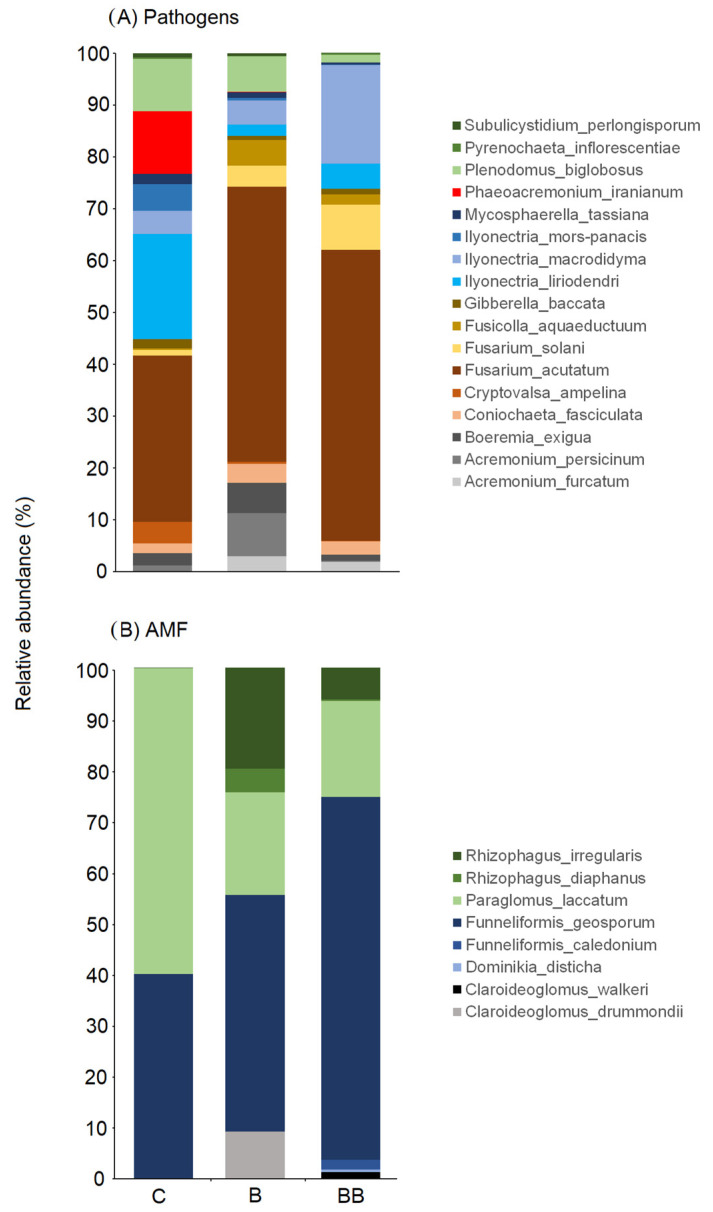
The relative abundance of various soil-borne fungal pathogens (**A**) and arbuscular mycorrhizal fungi (**B**) species in different soil treatments.

**Figure 3 microorganisms-12-00810-f003:**
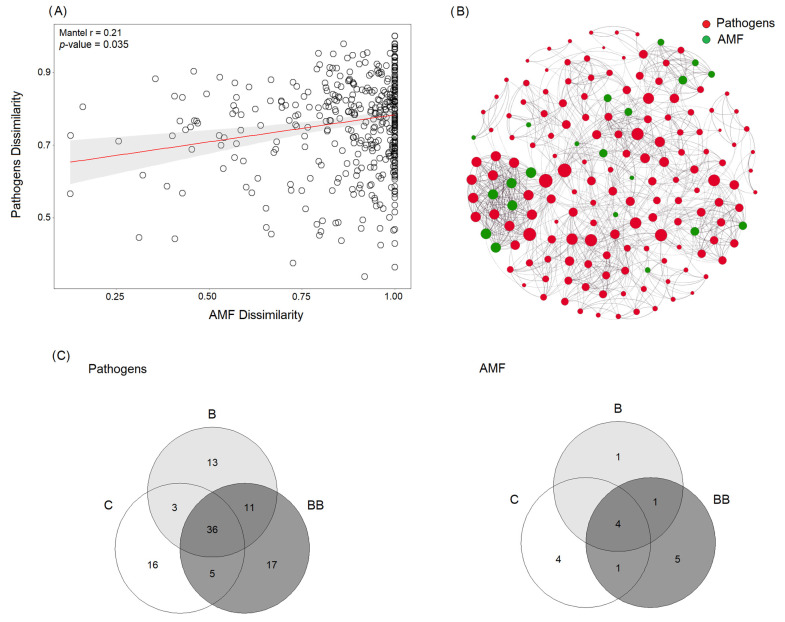
(**A**): Relationships between soil-borne fungal pathogens and AMF relative abundance using a Spearman correlation analysis. The relative abundance values on both axes are scaled from 0 to 1 for better visualization. (**B**): Correlation base network analysis showing potential interactions between soil-borne fungal pathogens and AMF. The connections indicate a strong (Spearman’s ρ > 0.6 and ρ < −0.6) and significant (*p*-value < 0.05) correlation. The size of each node is proportional to the species relative abundance. (**C**): Venn diagram showing the number of soil-borne fungal pathogens and AMF species shared or not shared by the studied treatments across the rhizosphere and bulk soil.

**Figure 4 microorganisms-12-00810-f004:**
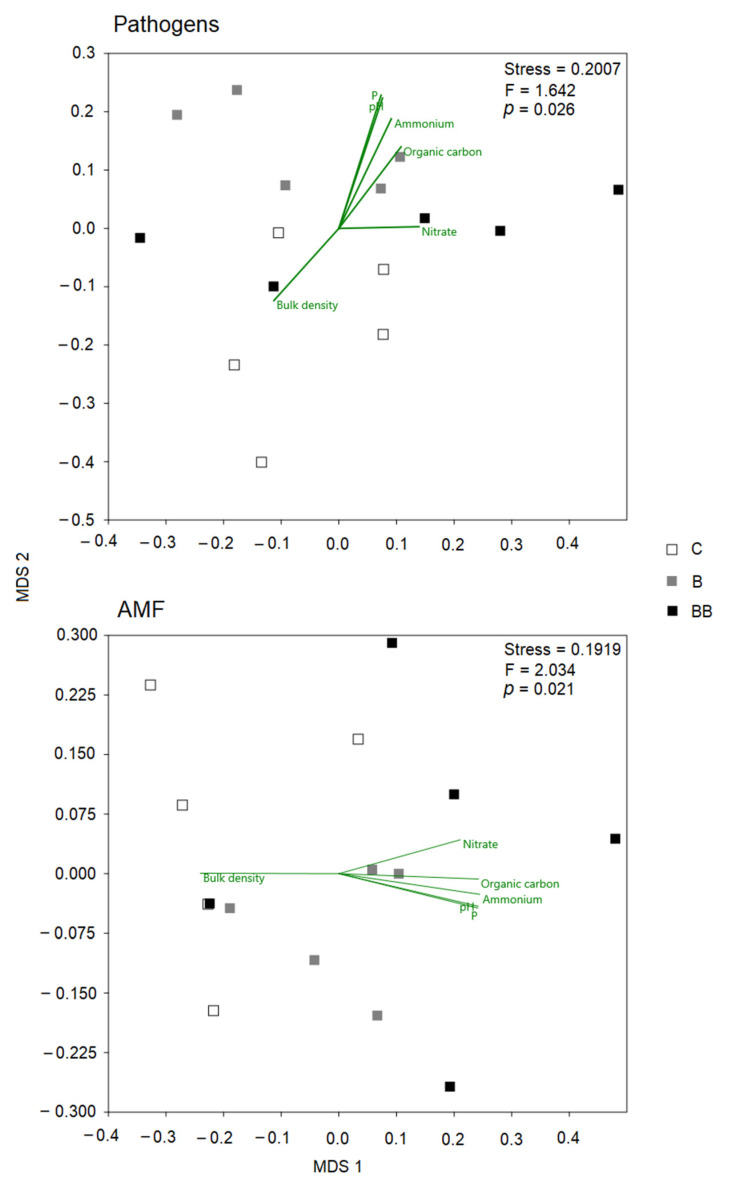
Nonmetric multidimensional scaling (NMDS) plots of soil-borne fungal pathogens and AMF communities in each of the three soil treatments across the rhizosphere soil. MDS axis 1 and MDS axis 2 represent the two axes of the two-dimensional ordination space. Each point represents the species community of one replicate of the plant. The stress level shown in each plot indicates how well the individual distances between objects are represented (between 0 and 1; the closer to 0, the better the original data points are represented in the ordination space). Vectors represent soil environmental variables that significantly correlated with the ordination (*p* < 0.05, based on 999 permutations).

**Table 1 microorganisms-12-00810-t001:** Chemical and physical characteristics of biochar applied in the conditioning field experiment (modified from Baronti et al. [38] and Genesio et al. [39]).

	Unit	Value
C	%	77.81
N	%	0.91
Al	mg·kg^−1^	268
C/N	-	63.53
Ca	mg·kg^−1^	25,000
Cu	mg·kg^−1^	97
Fe	mg·kg^−1^	333
K	mg·kg^−1^	13,900
Mg	mg·kg^−1^	28,700
Mn	mg·kg^−1^	84
Na	mg·kg^−1^	11,900
P	mg·kg^−1^	23,300
S	mg·kg^−1^	481
Zn	mg·kg^−1^	104
pH	-	9.8
CEC	Cmolc·kg^−1^	101
Max water absorption	gg^−1^ of d.m.	4.53
BET	m^2^·g^−1^	410 ± 6
Total porosity	mm^3^·g^−1^	2722
Transmission pores	mm^3^·g^−1^	318
Storage pores	mm^3^·g^−1^	1997
Residual pores	mm^3^·g^−1^	406
**Particle size distribution (mm):**		
50–20	%	4.45
20–10	%	12.1
10–8	%	13.1
8–4	%	10.36
4–2	%	19.85
2–1	%	24.2
<1	%	15.94

**Table 2 microorganisms-12-00810-t002:** Physical, chemical, and biochemical parameters in the untreated control (C), single biochar application (B), and double biochar application (BB). Different letters within each row indicate significant differences (Tukey test, *p* < 0.05).

Soil Parameters	Treatments		
	C Mean ± s.d.	B Mean ± s.d.	BB Mean ± s.d.
pH	6.33 ± 0.06 c	6.83 ± 0.11 b	7.07 ± 0.10 a
Bulk density (g·cm^−3^)	1.63 ± 0.03 a	1.59 ± 0.02 b	1.53 ± 0.02 c
Organic carbon (g·kg^−1^)	12.7 ± 0.67 c	17.3 ± 1.06 b	23.1 ± 1.15 a
Nitrate (NO^3−^-N (mg·kg^−1^))	1.51 ± 0.16 b	1.78 ± 0.12 b	5.88 ± 0.66 a
Ammonium (NH_4_^+^-N (mg·kg^−1^))	12.2 ± 0.42 b	13.4 ± 0.30 a	14.3 ± 1,29 a
P (mg·kg^−1^)	147.4 ± 12.1 c	262 ± 32.2 b	313 ± 31.4 a

## Data Availability

Data are available upon request from corresponding author.

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
