# Peer review of "Long-Term Application of Biochar Mitigates Negative Plant–Soil Feedback by Shaping Arbuscular Mycorrhizal Fungi and Fungal Pathogens"

_microorganisms, 2024, doi:10.3390/microorganisms12040810_

Round 1
Reviewer 1 Report
Comments and Suggestions for Authors
The paper entitled “Long-term application of biochar mitigates negative grapevine plant-soil feedback by shaping arbuscular mycorrhizal fungi and fungal pathogens” is an interesting work on a topic still partially unknown. The topic is clearly stated and the literature cited is sufficient and up-to-date. The materials and methods are correct and updated. The results are consistent with the analyzes and discussed appropriately. The conclusions are not speculative. The experimentation conducted is very rigorous and the text is excellently written, so I struggle to find sections that need improvement. The only uncertainty concerns figure 1, due to the non-homogeneous standard deviation for the grapevine control compared to all the other cases
Here my few comments:
L29: Please explain the AMF acronym
L128-133: I found no information regarding the cultivar and the type of rootstock. In particular, it is important to know if these combinations are homogeneous in the plots, also in terms of clones. The Montepulciano area suggests the use of the "Sangiovese" vine, but the high polyclonality of the cultivar and the profound differences between the different clones available are known.
155-158: I seem to have understood that the analysis of the land conditioned for 10 years was only done at the end of the period. In other words, wasn't a pre-evaluation of the microbiological characteristics (in particular of pathogens) of the soils from the 15 plots carried out before the treatments, in such a way as to guarantee a common starting point?
L171: Why the choice of this cultivar? In reference to the origin of the soil, I would have imagined Sangiovese. Alternatively, it would still make sense for the cultivar to be the same as the one from which the conditioned soil comes.
L236: As previously said, my only doubt concerns the initial analyses, that is, knowing how much the 3 theses have changed compared to the beginning, not just the differences between the theses after 10 years.
Fig.1. How do you explain the low variability of control in grape compared to other plants? In the treatment tests, the standard deviation is quite large (and similar) for all plants, including grapes. The controls also show in all other cases, deviations similar to those of the treatment tests, except for grape.
Author Response
The paper entitled “Long-term application of biochar mitigates negative grapevine plant-soil feedback by shaping arbuscular mycorrhizal fungi and fungal pathogens” is an interesting work on a topic still partially unknown. The topic is clearly stated, and the literature cited is sufficient and up-to-date. The materials and methods are correct and updated. The results are consistent with the analyzes and discussed appropriately. The conclusions are not speculative. The experimentation conducted is very rigorous and the text is excellently written, so I struggle to find sections that need improvement. The only uncertainty concerns figure 1, due to the non-homogeneous standard deviation for the grapevine control compared to all the other cases.
Response: We would like to thank the reviewer for acknowledging the efforts put in our work and the quality of the experimental design and the rest of the manuscript. Regarding the uncertainty presented by the reviewer, we would like to point out that although the figures were reassembled in one, the standard deviation and the statistical analysis were performed on each plant species individually. To avoid confusion, the figure’s caption was adjusted as follows: “The error bars represent the standard deviation for each plant species case. Bars topped by the same letter do not differ significantly by Tukey test on each plant species” Lines 289-291.
L29: Please explain the AMF acronym
Done. Line 27.
L128-133: I found no information regarding the cultivar and the type of rootstock. In particular, it is important to know if these combinations are homogeneous in the plots, also in terms of clones. The Montepulciano area suggests the use of the "Sangiovese" vine, but the high polyclonality of the cultivar and the profound differences between the different clones available are known.
Response: We agree with the reviewer that not mentioning the cultivar and rootstock type would open speculations on the lack of homogeneity in our plot experiment due to the presence of different available clones in the area, and for this reason the text was adjusted including this specific information as follows: “The vineyard (cv. Merlot, clone 181; rootstock 3309 Couderc) was established in 1995, and features rows of plants-oriented East-West …” Lines 146-147.
155-158: I seem to have understood that the analysis of the land conditioned for 10 years was only done at the end of the period. In other words, wasn't a pre-evaluation of the microbiological characteristics (in particular of pathogens) of the soils from the 15 plots carried out before the treatments, in such a way as to guarantee a common starting point?
Response: Right, the microbiome analysis took place a decade after the experiment commenced. Initially, the experiment was undertaken to assess the impact of biochar on water retention and grape yield within the Vineyard. As the experiment progressed, the opportunity arose to delve into the long-term effects of biochar on soil microbiome within the framework of the plant-soil feedback process. This decision stemmed from the realization that we had ten years' worth of conditioned soil readily available for analysis, prompting us to explore the broader ecological implications of our findings.
L171: Why the choice of this cultivar? In reference to the origin of the soil, I would have imagined Sangiovese. Alternatively, it would still make sense for the cultivar to be the same as the one from which the conditioned soil comes.
Response: Following the previous response, the reviewer would notice that we did use the same cultivar as the one used during the conditioning phase.
L236: As previously said, my only doubt concerns the initial analyses, that is, knowing how much the 3 theses have changed compared to the beginning, not just the differences between the theses after 10 years.
Response: We wish to inform the reviewer that the soil chemical properties at the commencement of the experiment have been referenced in the manuscript, as they were previously conducted (please refer to Baronti et al. [38] for further details). However, regrettably, the microbiological analyses were solely carried out at the experiment's conclusion. Our objective was to explore the impact of a decade-long conditioning on plant growth of conspecifics and heterospecifics by assessing alterations in both beneficial and detrimental soil fungi.
Fig.1. How do you explain the low variability of control in grape compared to other plants? In the treatment tests, the standard deviation is quite large (and similar) for all plants, including grapes. The controls also show in all other cases, deviations similar to those of the treatment tests, except for grape.
Response: We appreciate the reviewer's observation regarding the variability of the control group in grape plants compared to other species. We want to clarify to the reviewer that in our calculations, we utilized the shoot biomass replicates to determine the standard deviation. Consequently, a low standard deviation indicates that these replicates exhibited greater similarity among themselves compared to the replicates of other treatments. It's important to note that we didn't influence the variability observed in the other treatments. Our primary focus lies in discerning the statistical disparities between treatments rather than the standard deviation among them. This approach allows us to pinpoint meaningful differences in the experimental conditions rather than variations within each treatment group.
Reviewer 2 Report
Comments and Suggestions for Authors
Overall, the work is well-structured, this contribution should be considered for publication after addressing the following comments
1. The abstract should be modified, Specify the purpose or objective of the study to provide context for readers and explain the significance of incorporating biochar into the experiment.
2. The last part of the introduction must improve and consider expanding on the potential benefits of biochar application in alleviating negative PSF for V. vinifera. What specific mechanisms or processes are involved in biochar's ability to mitigate negative PSF? Ensure clarity and precision in your language to effectively communicate your hypotheses. Avoid vague or ambiguous terms that may lead to confusion or misinterpretation.
3. In Section 2 Materials and Methods provide more context regarding the long-term field experiment from which the soil samples were obtained for the conditioning phase. What were the key treatments or factors manipulated in this field experiment?
4. In section 2.4 Data analysis, Provide a brief overview of the STATISTICA 13.3 software.
5. In section 3.2 Clarify the methodology used to measure shoot biomass and how it was analyzed statistically Please explain in the text.
6. Consider revising the conclusion to highlight the broader implications of your findings and potential avenues for future research. What are the practical applications of your results, and what questions remain unanswered?
Comments on the Quality of English LanguageThe manuscript is well written, with few suggestions for changes, but it is important to follow the reviewer's suggestions. Important and very current topic.
Author Response
Overall, the work is well-structured, this contribution should be considered for publication after addressing the following comments.
Response: Thank you for the positive feedback and for the valuable comments, which were considered seriously for improving our manuscript.
The abstract should be modified, Specify the purpose or objective of the study to provide context for readers and explain the significance of incorporating biochar into the experiment.
Done. The abstract background section was adjusted as follows: “Negative plant-soil feedback (PSF) arises when localized accumulations of pathogens reduce the growth of conspecifics, whereas positive PSF can occur due to the emergence of mutualists. Biochar, a carbon-rich material produced by the pyrolysis of organic matter, has been shown to modulate soil microbial communities by altering their abundance, diversity, and activity. For this reason, to assess the long-term impact of biochar on soil microbiome dynamics and subsequent plant performance, we conducted a PSF greenhouse experiment using field soil conditioned over 10 years with Vitis vinifera (L.), without (e.g., C) or with biochar at two rates (e.g., B and BB).” Lines 17-24.
The last part of the introduction must improve and consider expanding on the potential benefits of biochar application in alleviating negative PSF for V. vinifera. What specific mechanisms or processes are involved in biochar's ability to mitigate negative PSF? Ensure clarity and precision in your language to effectively communicate your hypotheses. Avoid vague or ambiguous terms that may lead to confusion or misinterpretation.
Response: We would like to inform the reviewer that the comment idea was already established in the introduction, starting from stating the impact of biochar on bacterial and fungal soil communities, and then on the beneficial and pathogenic communities as follows: “Additionally, biochar is recognized for its capacity to reshape soil bacterial community structure and diversity [30, 31]. Furthermore, investigations have probed the influence of biochar on fungal abundance and diversity, revealing a range of outcomes, including fluctuations in the colonization and abundance of AMF [32,33], diminished fungal diversity attributed to the inability of certain fungal taxa to adapt to rapid soil environmental changes [31], and a reduction in fungal abundance [34]. Moreover, biochar has been observed to mitigate disease incidence and severity caused by fungal pathogens, largely attributable to its modification of soil microbiota and induction of plant systemic resistance [35,36]. Therefore, it could be hypothesized that biochar application can be considered as an ecological practice to mitigate the negative PSF caused by soilborne pathogens in agroecosystems.” Lines 89-100, and then at the last paragraph we delve into the proposed mechanism wherein biochar mitigates the negative PSF of Vitis. This mechanism involves the modulation of soil microbiota, as previously established, by diminishing pathogens and bolstering mutualists: “In addition, we hypothesized that the application of biochar would help alleviate the negative PSF for V. vinifera throughout its well-established effects on soil microbial communities, probably by reducing fungal pathogens and enhancing the AMF community in the soil.” 126-129.
In Section 2 Materials and Methods provide more context regarding the long-term field experiment from which the soil samples were obtained for the conditioning phase. What were the key treatments or factors manipulated in this field experiment?
Response: We appreciate the reviewer's feedback. We want to highlight that in Section 2 of the Materials and Methods, we have provided a comprehensive overview of the long-term field experiment conducted at the 'La Braccesca Estate' vineyard in Montepulciano. We have detailed the key treatments applied in the experiment, including single and double biochar applications, alongside the control plots. Moreover, we've described the vineyard's characteristics, such as its location, grapevine variety, cultivation practices, soil properties, and the specifics of the biochar utilized. If there are additional details or aspects the reviewer would like to be emphasized, please do not hesitate to specify, and we will ensure to address them accordingly.
In section 2.4 Data analysis, Provide a brief overview of the STATISTICA 13.3 software.
Done. The text was adjusted as follows: “All statistical analyses were performed using STATISTICA 13.3 software (TIBCO Software Inc., Palo Alto, CA, USA) which offers a comprehensive suite of tools for data exploration, visualization, and hypothesis testing.” Lines 241-243.
In section 3.2 Clarify the methodology used to measure shoot biomass and how it was analyzed statistically Please explain in the text.
Response: We would like to inform the reviewer that Section 3 generally pertains to results description rather than methodology explanation, hence our focus on providing a simple description of our results. However, we want to assure the reviewer that the comments raised have already been addressed in detail within the Materials and Methods sections. Regarding the methodology used for shoot biomass measurement, we quote from the manuscript: “Plants were cut at ground level, and the shoots were subjected to drying at 70 °C for 72 hours, after which their dry weight was recorded.” (Lines 199-201). Furthermore, regarding the statistical analysis approach, we refer the reviewer to the manuscript: “The statistical significance of the biomass data obtained from the experiment was evaluated using two-way ANOVA (analysis of variance) to determine the main and interactive effects of the fixed factors of conditioning treatment and plant species on shoot biomass. The results of the analysis of variance were submitted by pairwise Tukey test comparing the individual means of response plants in each soil treatment. The level of significant differences was evaluated with p < 0.05.” (Lines 233-238).
Consider revising the conclusion to highlight the broader implications of your findings and potential avenues for future research. What are the practical applications of your results, and what questions remain unanswered?
Done. The conclusions section was totally modified following the reviewer’s suggestion: “Our study, investigating for the first time 10 years of conditioning with biochar at two different rates, offers valuable insights into the potential of biochar as a sustainable soil management tool in agricultural ecosystems. By conducting a feedback experiment over a decade-long period, we demonstrated that biochar amendment can effectively mitigate the negative PSF effects associated with continuous cropping of V. vinifera. Our results not only underscore the importance of biochar in enhancing plant growth but also shed light on its role in modulating soil microbial communities, particularly AMF and fungal pathogens. The practical applications of our findings extend beyond improving crop productivity to encompass environmental sustainability and resilience in agricultural systems. By reducing the relative abundance of specialized fungal pathogens and enhancing the diversity of beneficial AMF, biochar application can contribute to the development of more resilient agroecosystems. Furthermore, the absence of antagonism between soil communities suggests that biochar promotes a balanced soil microbiome conducive to plant growth. However, several questions re-main unanswered, presenting avenues for future research. Firstly, further investigation is warranted to elucidate the underlying mechanisms through which biochar interacts with soil microbial communities and influences plant-soil interactions. Additionally, long-term field studies are needed to assess the scalability and durability of biochar-mediated improvements in soil health and crop performance across diverse agroecological contexts. Moreover, exploring the potential synergies between biochar and other soil amendments or management practices could enhance our understanding of integrated soil fertility management strategies.” Lines 460-480.
The manuscript is well written, with few suggestions for changes, but it is important to follow the reviewer's suggestions. Important and very current topic.
Response: Thank you for acknowledging the novelty and importance of our work.
Reviewer 3 Report
Comments and Suggestions for Authors
The MS, despite covering a very interesting topic, is not sufficiently well organized/written.
Indeed:
(a) Both Abstract and Conclusions lack both numerical or clear indices regarding the results obtained and indications / comments regarding their novelty.
b) In M&Ms the characteristics of biochar are summarily reported, it would be appropriate to report the reference table published in ref. [39].
c) Regarding the "response phase" the authors should specify how the soil sampling was carried out, at what depth interval, removing or not a surface layer, from which or from all 5 replicates of treated soil, and then the mixing of the different samples should be explained, if any.
(d) Line 173: The words "all plants survived" appear improper in the absence of indication of the mode of cultivation (greenhouse, outdoors, with what frequency of irrigation, fertigation if any, etc.?).
(e) Since a negative effect is hypothesized, this also should be demonstrated by introducing as additional control the soil before biochar treatment (or the uncultivated soil); also, a comparison between biochar and organic matter administration with a different material would have been appropriate.
f) Comparing with previous publications by the same authors [38, 39] an increase in available soil water content or increased plant water availability is no longer evaluated, why?
g) Also, it will be useful to indicate in Results which of the microbial species are not vine pathogens.
(h) Section 3.4: The correlation between AMF and pH should be discussed considering that the pH of biochar is 9.8 and results in an increase in pH from sample C to BB. Indeed, it can be hypothesized that it is the pH that determines a higher or lower presence of pathogenic microorganisms.
(i) formal errors the first of which is rather relevant. In Table 1 the k of kg is incorrectly capitalized in three cases; V. vinifera on lines 156, 262 and 417 is not written in italics, which is necessary to check for all other species mentioned in the text; the acronym AMF is not explained at its first introduction.
Author Response
The MS, despite covering a very interesting topic, is not sufficiently well organized/written.
Response: We want to show our gratitude to the reviewer for the constructive comments, which has helped us in improving the quality of our manuscript.
(a) Both Abstract and Conclusions lack both numerical or clear indices regarding the results obtained and indications / comments regarding their novelty.
Response: We agree with the reviewer and thus we changed the Abstract and Conclusion sections to include the missing sentences as follows: “Negative plant-soil feedback (PSF) arises when localized accumulations of pathogens reduce the growth of conspecifics, whereas positive PSF can occur due to the emergence of mutualists. Biochar, a carbon-rich material produced by the pyrolysis of organic matter, has been shown to modulate soil microbial communities by altering their abundance, diversity, and activity. For this reason, to assess the long-term impact of biochar on soil microbiome dynamics and subsequent plant performance, we conducted a PSF greenhouse experiment using field soil conditioned over 10 years with Vitis vinifera (L.), without (e.g., C) or with biochar at two rates (e.g., B and BB). Subsequently, the conditioned soil was employed in a response phase involving either the same plant species or different species, i.e., Medicago sativa (L.), Lolium perenne (L.), and Solanum lycopersicum (L.). We utilized next-generation sequencing to assess the abundance and diversity of fungal pathogens and arbuscular mycorrhizal fungi (AMF) within each conditioned soil. Our findings demonstrate that biochar application exerted a stimulatory effect on the growth of both conspecifics and heterospecifics. In addition, our results show that untreated soils had higher abundance of grape specialized fungal pathogens, mainly Ilyonectria liriodendra, with a relative abundance of 20.6% compared to 2.1% and 5.1% in B and BB, respectively. Cryptovalsa ampelina also demonstrated higher prevalence in untreated soils, accounting for 4.3% compared to 0.4% in B and 0.1% in BB. Additionally, Phaeoacremonium iranianum was exclusively present in untreated soils, comprising 12.2% of the pathogens’ population. Conversely, the application of biochar reduced generalist fungal pathogens. For instance, Plenodomus biglobosus decreased from 10.5% in C to 7.1% in B and 2.3% in BB, while Ilyonectria mors-panacis declined from 5.8% in C to 0.5% in B and 0.2% in BB. Furthermore, biochar application was found to enrich the AMF community. Notably, certain species like Funneliformis geosporum exhibited increased relative abundance in biochar-treated soils, reaching 46.8% in B and 70.3% in BB, compared to 40.5% in untreated soils. Concurrently, other AMF species, namely Rhizophagus irregularis, Rhizophagus diaphanus, and Claroideoglomus drummondii, were exclusively observed in soils where biochar was applied. We propose that the alleviation of negative PSF can be attributed to the positive influence of AMF in the absence of strong inhibition by pathogens. In conclusion, our study underscores the potential of biochar application as a strategic agricultural practice for promoting sustainable soil management over the long term” Lines 17-44.
And: “Our study, investigating for the first time 10 years of conditioning with biochar at two different rates, offers valuable insights into the potential of biochar as a sustainable soil management tool in agricultural ecosystems. By conducting a feedback experiment over a decade-long period, we demonstrated that biochar amendment can effectively mitigate the negative PSF effects associated with continuous cropping of V. vinifera. Our results not only underscore the importance of biochar in enhancing plant growth but also shed light on its role in modulating soil microbial communities, particularly AMF and fungal pathogens. The practical applications of our findings extend beyond improving crop productivity to encompass environmental sustainability and resilience in agricultural systems. By reducing the relative abundance of specialized fungal pathogens and enhancing the diversity of beneficial AMF, biochar application can contribute to the development of more resilient agroecosystems. Furthermore, the absence of antagonism between soil communities suggests that biochar promotes a balanced soil microbiome conducive to plant growth. However, several questions remain unanswered, presenting avenues for future research. Firstly, further investigation is warranted to elucidate the underlying mechanisms through which biochar interacts with soil microbial communities and influences plant-soil interactions. Additionally, long-term field studies are needed to assess the scalability and durability of biochar-mediated improvements in soil health and crop performance across diverse agroecological contexts. Moreover, exploring the potential synergies between biochar and other soil amendments or management practices could enhance our understanding of integrated soil fertility management strategies.” Lines 460-480.
b) In M&Ms the characteristics of biochar are summarily reported, it would be appropriate to report the reference table published in ref. [39].
Done. Following the reviewer’s suggestion, the table was reported in the manuscript as new Table 1.
c) Regarding the "response phase" the authors should specify how the soil sampling was carried out, at what depth interval, removing or not a surface layer, from which or from all 5 replicates of treated soil, and then the mixing of the different samples should be explained, if any.
Response: We would like to inform the reviewer that soil sampling was performed during the “conditioning phase” not the “response phase”. Assuming the reviewer intended the conditioning phase, the section was adjusted as follows: “After 10 years, soil samples for the response phase were collected for each treatment from three types of conditioned soils: soils conditioned with V. vinifera without treatment, with the application of biochar in one rate, and with the application of biochar in two rates. For further analysis, soil samples were collected using a 10 cm diameter soil corer, reaching a depth of 0–20 cm after the removal of above-ground litter. This was performed at three distinct points within each replicate plot to create a composite sample representative of each plot. A total of five replicate samples were collected for each treatment, resulting in 15 samples overall. Subsequently, soil samples were processed in the laboratory. Initially, they were sieved at field moisture using a 2 mm mesh. Following sieving, the samples were divided into three portions: one portion was stored at 4 °C for the analysis of soil biochemical activities, another portion was preserved at −80 °C for DNA extraction, and the remaining portion was air-dried for subsequent chemical analysis. The chemical, biochemical, and metagenomics properties of each of the conditioned soils are described in Idbella et al. [40].” Lines 167-180.
(d) Line 173: The words "all plants survived" appear improper in the absence of indication of the mode of cultivation (greenhouse, outdoors, with what frequency of irrigation, fertigation if any, etc.?).
Done. The paragraph was adjusted as follows: “In this phase, conducted in a greenhouse of the Department of Agriculture (Federico II University of Naples, Italy), the pots (20 cm opening diameter ∗ 18 cm height ∗ 15 cm bottom diameter) were filled with the previously conditioned soils, which included soil with a single biochar application, soil with two biochar applications, and control without biochar. Each treatment, as well as each of the tested plant species, had five replicates, resulting in a total of 60 pots (4 plant species x 3 treatments x 5 replicates). For each heterospecific plant species, ten seeds were sown in each pot. These seeds underwent surface sterilization in a 3% sodium hypochlorite solution for 1 minute and were thoroughly rinsed with sterile water before use. After germination, the number of seedlings in each pot was reduced to five. In the case of V. vinifera, two seedlings of the same variety (cv. Merlot, clone 181; rootstock 3309 Couderc) were placed in each corresponding pot. Pots were arranged in a randomized block design, and each pot was located inside an individual pot saucer in order to avoid contamination among plants through leaching or splashing when watering. Pots were watered to field capacity, when necessary, by adding distilled water on the individual container. Following the response phase, which lasted for 100 days for S. lycopersicum, 70 days for M. sativa, 120 days for L. perenne, and 230 days for V. vinifera, all plants survived and were harvested. Plants were cut at ground level, and the shoots were subjected to drying at 70 °C for 72 hours, after which their dry weight was recorded.” Lines 183-201.
(e) Since a negative effect is hypothesized, this also should be demonstrated by introducing as additional control the soil before biochar treatment (or the uncultivated soil); also, a comparison between biochar and organic matter administration with a different material would have been appropriate.
Response: We acknowledge the reviewer's suggestion regarding the inclusion of uncultivated soil in our experimentation. While we recognize the merit of such an addition, our primary objective was to assess the impact of biochar on cultivated soil over the long term. Specifically, we aimed to investigate whether the incorporation of biochar would yield positive or negative effects within this specific context. Additionally, within the framework of plant-soil feedback, the use of uncultivated soil would not align with our goal of observing how the soil responds to conditioning by plants, particularly in the presence or absence of biochar. Furthermore, we appreciate the reviewer's insight regarding the comparison between biochar and various organic amendments. Although we agree that such a comparison would be valuable, it's worth noting that our experiment initially focused on evaluating the effects of biochar on water retention. The decision to incorporate the concept of plant-soil feedback emerged later in the research process.
f) Comparing with previous publications by the same authors [38, 39] an increase in available soil water content or increased plant water availability is no longer evaluated, why?
Response: We opted not to focus on soil water content in our plant-soil feedback experiment since water availability wasn't a limiting factor. Throughout the experiment, the soil was consistently irrigated to field capacity. However, we did not introduce additional soil nutrients, which were of primary interest in our study along with the fungal community.
g) Also, it will be useful to indicate in Results which of the microbial species are not vine pathogens.
Done. The results section was adjusted as follows: “At the species level, fungal pathogens and AMF communities differed between treatments (Fig. 2). Specifically, the control soil (C) was characterized by higher abundance of generalist pathogens such as Plenodomus biglobosus with 10.5% compared to 7.1% in B and 2.3% in BB, and Ilyonectria mors-panacis with 5.8% compared to 0.5% in B and 0.2% in BB, as well as higher abundance of grape specialized pathogens such as Ilyonectria liriodendra with 20.6% compared to 2.1% in B and 5.1% in BB, Cryptovalsa ampelina with 4.3% compared to 0.4% in B and 0.1% in BB, and Phaeoacremonium iranianum that was present only in treatment C with 12.2%, while it was absent in the other treatments. On the other hand, other generalist pathogens like Ilyonectria macrodidyma and Fusarium solani were more abundant in BB with 19.9% and 9.7%, respectively, compared to 4.6% and 1.3% in C and 5.6% and 4.4% in B, respectively. Fusarium acuta-tum, however, was abundant in all soils, but was more common in BB (56.2%), followed by B (53.5%), while it accounted for 32.7% in the control. Boeremia exigua, Acremonium persicinum and Acremonium furcatum were more abundant in soil B with 6.3%, 8.9% and 3.8% than in soil C with 2.6%, 1.0% and 0% and in BB with 1.1%, 0.1% and 2.5%, respectively.” Lines 294-308.
(h) Section 3.4: The correlation between AMF and pH should be discussed considering that the pH of biochar is 9.8 and results in an increase in pH from sample C to BB. Indeed, it can be hypothesized that it is the pH that determines a higher or lower presence of pathogenic microorganisms.
Done. The paragraph was added in the Discussion section as follows: “Our nMDS analysis unveiled a multifaceted interaction among parameters shaping the distribution of fungal communities. Within both B and BB soils, a positive correlation was observed between fungi and parameters such as P, pH, OC, ammonium, and nitrate concentrations. These associations imply that the incorporation of biochar into soils fosters conditions conducive to fungal proliferation, characterized by enhanced nutrient availability, pH levels, and increased carbon content [87-88]. Notably, soil pH serves as a reflection of soil nutrient availability and plays a pivotal role in regulating the accessibility of other nutrients via ion exchange processes [89]. Fluctuations in soil pH can precipitate alterations in species richness and community composition. Specifically, soil pH exerts significant influence on the physiological dynamics of AMF, affecting crucial aspects such as sporulation [90] and hyphal growth [91]. Therefore, the observed shifts in AMF physiology induced by changes in soil pH, facilitated by biochar application, may contribute to the reduction of fungal pathogen populations in agricultural soils. This underscores the multifaceted implications of soil pH modulation through biochar amendments, not only on fungal community dynamics but also on plant health and disease suppression in agroecosystems.” Lines 443-458.
(i) formal errors the first of which is rather relevant. In Table 1 the k of kg is incorrectly capitalized in three cases; V. vinifera on lines 156, 262 and 417 is not written in italics, which is necessary to check for all other species mentioned in the text; the acronym AMF is not explained at its first introduction.
Response: We would like to inform the reviewer that the comments were adjusted in the manuscript, please refer to the track-version manuscript for more details.
Round 2
Reviewer 3 Report
Comments and Suggestions for Authors
The MS was revised taking into account previous comments with a good result.
One final request remains, however: please add a comment regarding the increase in soil pH from 6.33 to 7.07 in case of BB, referring to the pH range considered the optimum for grapes and the pH range within vines grow without problems.
Author Response
The MS was revised taking into account previous comments with a good result.
Response: We would like to thank the reviewer for his serious valuable comments, which indeed have helped in improving the quality of our manuscript, and for recognizing the efforts made to respond to such comments.
One final request remains, however: please add a comment regarding the increase in soil pH from 6.33 to 7.07 in case of BB, referring to the pH range considered the optimum for grapes and the pH range within vines grow without problems.
Done. The suggested section was added in the Discussion section as follows: "The long-term increase in soil pH values from 6.33 in C to 6.83 in B and 7.07 in BB, observed ten years after biochar application, persists even under conventional vineyard management. This elevation may be attributed not solely to the initially high pH value (9.8) of the biochar utilized [39], but also to the presence of base cations in the soil and the enduring effects of biochar basic oxides [90]. Additional mechanisms contributing to pH elevation could involve processes associated with plant-mediated NO3-N sorption, which can generate OH− ions and subsequently raise pH levels. This phenomenon contrasts with NH4+ uptake, which yields H+ ions, thereby contributing to soil acidification [91]. capillary forces within biochar micropores possess the capacity to mitigate the acidifying potential of such ions within the soil solution. Upon contact with biochar micropores, these ions may experience reduced diffusion and adsorption onto biochar surfaces due to capillary effects. Consequently, the immediate availability of ions for reactions leading to soil acidification is diminished. The observed increase in soil pH over the long term suggests improved conditions for grapevine cultivation, potentially enhancing nutrient uptake and water retention. Moreover, the mitigation of soil acidification by biochar may contribute to the sustainability of vineyard ecosystems, preserving soil fertility and minimizing the need for external amendments." Lines 450-466.